# Net Primary Productivity of *Pinus massoniana* Dependence on Climate, Soil and Forest Characteristics

**Xin Huang [1] , Chunbo Huang [2] , Mingjun Teng [1], Zhixiang Zhou [1],\* and Pengcheng Wang [1]**

1   College of Horticulture and Forestry Sciences/Hubei Engineering Technology Research Center for Forestry Information, Huazhong Agricultural University, Wuhan 430070, China; hxhanson@163.com (X.H.); tengmingjun@hotmail.com (M.T.); wangpc@mail.hzau.edu.cn (P.W.)
2   School of Geography and Information Engineering, China University of Geosciences, Wuhan 430074, China; poahcb@hotmail.com
\*   Correspondence: whzhouzx@mail.hzau.edu.cn; Tel.: +86-027-8728-4232

**Abstract:** Understanding the spatial variation of forest productivity and its driving factors on a large regional scale can help reveal the response mechanism of tree growth to climate change, and is an important prerequisite for efficient forest management and studying regional and global carbon cycles. *Pinus massoniana* Lamb. is a major planted tree species in southern China, playing an important role in the development of forestry due to its high economic and ecological benefits. Here, we establish a biomass database for *P. massoniana*, including stems, branches, leaves, roots, aboveground organs and total tree, by collecting the published literature, to increase our understanding of net primary productivity (NPP) geographical trends for each tree component and their influencing factors across the entire geographical distribution of the species in southern China. *P. massoniana* NPP ranges from 1.04 to 13.13 Mg·ha$^{-1}$·year$^{-1}$, with a mean value of 5.65 Mg·ha$^{-1}$·year$^{-1}$. The NPP of both tree components (i.e., stem, branch, leaf, root, aboveground organs, and total tree) show no clear relationships with longitude and elevation, but an inverse relationship with latitude ($p < 0.01$). Linear mixed-effects models (LMMs) are employed to analyze the effect of environmental factors and stand characteristics on *P. massoniana* NPP. LMM results reveal that the NPP of different tree components have different sensitivities to environmental and stand variables. Appropriate temperature and soil nutrients (particularly soil available phosphorus) are beneficial to biomass accumulation of this species. It is worth noting that the high temperature in July and August (HTWM) is a significant climate stressor across the species geographical distribution and is not restricted to marginal populations in the low latitude area. Temperature was a key environmental factor behind the inverse latitudinal trends of *P. massoniana* NPP, because it showed a higher sensitivity than other factors. In the context of climate warming and nitrogen (N) deposition, the inhibition effect caused by high temperatures and the lack or imbalance of soil nutrients, particularly soil phosphorus, should be paid more attention in the future. These findings advance our understanding about the factors influencing the productivity of each *P. massoniana* tree component across the full geographical distribution of the species, and are therefore valuable for forecasting climate-induced variation in forest productivity.

**Keywords:** net primary productivity; *Pinus massoniana*; geographical gradient; environmental factors; stand characteristics; regional scale

## 1. Introduction

Forest ecosystems are a major component of the terrestrial ecosystem worldwide and play an irreplaceable role in regulating global carbon balances and mitigating atmospheric concentrations of

greenhouse gases, as well as in biodiversity and water conservation [1,2]. Forest biomass is the total amount of organic matter formed by $CO_2$ sequestration in the process of photosynthesis per unit area, and its accumulation rate is usually used as an indicator of forest productivity [3]. Therefore, forest biomass and productivity are important for measuring carbon sequestration ability and assessing carbon balance of forest ecosystems, and play an important role in global carbon cycle research [4]. Since the industrial revolution in the mid-20th century, the rapid development of human activities and modern industry, especially the burning of fossil fuels, massive deforestation, and grassland reclamation, have had a tremendous impact on the global ecosystem [5]. Therefore, studies on the environmental response of forest productivity in the context of global climate change, such as atmospheric nitrogen (N) deposition, elevated $CO_2$ concentration, and climate warming, have received increasing attention because of their great practical significance for reducing carbon emissions and mitigating global warming [6,7].

The production, allocation, and turnover of forest carbon on a large regional scale has received considerable attention in the past and is becoming increasingly important [8]. The accurate assessment of forest productivity on a large regional scale and the understanding of its influencing factors can provide a theoretical basis for enhancing forest productivity and studying the terrestrial carbon cycle [9]. Because of the complex terrain and large environmental differences in China, the geographical distribution of forest biomass and productivity is diverse [10–12]. As an important component of forest resources in China, plantations play an increasingly prominent role in maintaining the global carbon balance and mitigating global warming [13]. As artificially regulated ecosystems, the carbon sequestration function of plantation ecosystems is directly or indirectly affected by tree species, afforestation strategies, and tending operations [14]. Most large-scale studies on biomass and productivity have been conducted for a variety of forest types in China, rather than for a particular tree species, although many of these forests are single-species plantations [15]. Research on the response of forest productivity to environmental factors at the tree species level can reveal species-specific ecophysiological characteristics. On this basis, more targeted forest management strategies (e.g., site selection for afforestation and selection of tree species for close-to-nature transformation of monospecific plantations) could be proposed for different tree species in different regions, especially for major planted tree species, to improve forest productivity and carbon storage [15,16].

In the past 20 years, China has implemented a series of ecological projects, such as the Grain-to-Green Program, to improve the situation of low forest coverage and poor forest resources, and has now become the country with the largest area of plantations [17]. *Pinus massoniana* Lamb. is widely distributed in subtropical areas of China. It is an important afforestation tree species in barren mountains owing to its strong adaptability, fast growth, high yield, and drought tolerance, and is also one of the most representative forest types in China [18]. As one of the major planted and native tree species in southern China, *P. massoniana* has made a great contribution to plantation development [13,15]. The eighth forest resources inventory of China published by the State Forestry Administration in 2014 showed that the total area and volume of *P. massoniana* were $1.0 \times 10^7$ ha and $5.9 \times 10^8$ m$^3$, respectively, accounting for 6.1% and 4.0% of the total area and volume of arbor forests in China. However, *P. massoniana* plantations are faced with the problems of their irrational structure, poor stability and degraded productivity. Therefore, seeking scientific ways of regulating its productivity to improve its ecological and economic benefits is a major concern. Studies on biomass and productivity of *P. massoniana* began in the early 1980s [19], and mainly focused on biomass estimation and its allocation patterns in different site conditions, stand ages and densities [20], allometric equations research [21] and the effect of species mixtures on biomass [22]. In general, the research objectives of the above-mentioned studies mainly focused on quantitative assessment of biomass, and few studies involved biomass accumulation rate, which is usually quantified by forest productivity [23]. Therefore, knowledge about productivity responses to environmental factors is underdeveloped. Moreover, although tree-ring chronologies have been widely used to explore the relationship between climate and *P. massoniana* tree growth [24], these experiments have been carried out in individual study sites on a local scale,

and the variation in productivity and its environmental control on a regional scale is still unclear. In addition, biotic factors, such as tree age and density, also influence forest productivity [25]. The variable environmental factors (e.g., temperature, precipitation, and soil properties) and forest characteristics (e.g., stand age and density) within the *P. massoniana* distribution area offers an opportunity to examine its productivity distribution patterns and its influencing factors on a large regional scale.

Numerous quantitative evaluations in the form of meta-analyses have been carried out to explore forest productivity and its influencing factors [26,27]. A lot of field measurements of *P. massoniana* biomass and productivity estimations have been conducted at multiple sites over the past four decades. However, these data were mostly published in Chinese journals and reports and are not accessible to non-Chinese scientists [27]. A *P. massoniana* biomass database could promote the completion of the established biomass database for China's forests and contribute to generating a huge database worldwide that will aid in the validation of ecosystem models, improving our understanding of the global carbon cycle and accurately evaluating carbon storage [26].

In this study, we established a biomass database for *P. massoniana*, including stems, branches, leaves, roots, aboveground organs, and total tree values by collecting data from the published literature, to increase our understanding of the geographical trends of *P. massoniana* net primary productivity (NPP) and its influencing factors across the full geographical distribution of the species in southern China. Our objectives were: (1) to document the spatial distribution of *P. massoniana* NPP on a large regional scale, (2) to quantify the effects of environmental factors (climate and soil variables) and stand characteristics on NPP of each tree component (i.e., stem, branch, leaf, root, aboveground organs and total tree), and (3) to identify the key environmental factors causing spatial heterogeneity in *P. massoniana* NPP.

## 2. Materials and Methods

### 2.1. Data Collection and Treatments

Our *P. massoniana* biomass database was generated by collecting published data (Figure 1, Table S1). To compile a comprehensive database, we searched the Web of Science for English literature and China National Knowledge Internet for Chinese literature. To minimize the variability associated with comparing biomass/productivity estimates derived from different methodologies, we selected all references included in the database that (1) included data actually measured in field experiments on *P. massoniana* stands (not forest inventory data or remote sensing based studies), (2) did not contain severe anthropogenic disturbances, such as close-to-nature transformation, irrigation, and harvest and (3) only included measurements of monospecific stands of *P. massoniana*; data on mixed stands was excluded. *P. massoniana* growth is affected by competition with other tree species in mixed stands [22]. The different types of species interactions formed depending on tree species composition, stand age, afforestation density and site conditions bring uncertainties to the evaluation of functions and services of forest ecosystems, including productivity [28]. Moreover, we only selected studies that were (4) derived from stable growing communities (data from stands younger than three years old was excluded), and (5) contained at least one of the following biomass data at the stand level: stem, branch, leaf, root, sum of aboveground organs or total tree. Studies on allometric scaling among biomass components of *P. massoniana* have focused primarily on the individual tree level [21]. However, forest NPP can only be estimated at the stand level [29]. All biomass data were converted to common units (Mg·ha$^{-1}$) prior to analysis. We retrieved missing latitude or longitude information for 16 sites without such data from Google Earth according to the site name [27]. Elevation information not provided in the literature was obtained based on the longitude, latitude of the sampling sites and a 1 km resolution Digital Elevation Model obtained from Cold and Arid Regions Sciences Data Center at Lanzhou (http://westdc.westgis.ac.cn/). A total of 87 references were acquired after screening (Figure 1, Table S1).

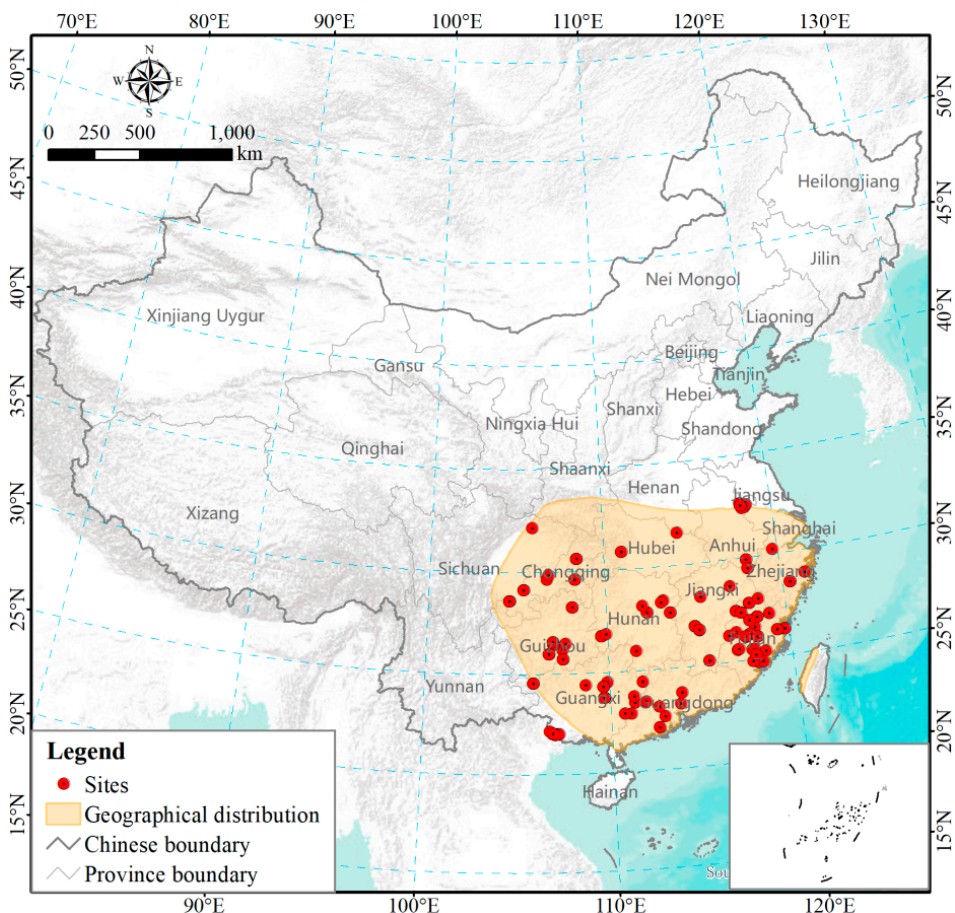

**Figure 1.** Geographical distribution of field study sites included in our database. Some plots are not visible as they are very close to each other and overlap. Geographical distribution range (shaded area) of *P. massoniana* adapted from Zhou (2001) [18].

Most of the literature only include biomass data and NPP data was rarely provided. Therefore, a uniform formula was used to estimate productivity based on biomass and stand age to enhance data comparability between different studies. NPP is the rate of production of biomass and organic compounds by the plant or ecosystem and consists of three components [23]:

$$NPP = \Delta Y + \Delta L + \Delta G, \qquad (1)$$

where $\Delta Y$ is the growth increment in a specified time interval (usually 1 year) estimated from temporal changes in forest biomass, $\Delta L$ is the loss part, consists of the death and litterfall production, and $\Delta G$ is the rate of grazing by herbivores. The full suite of components of NPP is rarely measured in forest ecosystems, owing to the difficulty of measuring $\Delta L$ and $\Delta G$. Considering the large amount of missing data for these two components in the literature, the NPP of total tree ($NPP_{tree}$) includes aboveground and belowground NPP and is the sum of four compartments in this study:

$$NPP_{tree} = NPP_{stem} + NPP_{bra} + NPP_{leaf} + NPP_{root}, \qquad (2)$$

where $NPP_{stem}$, $NPP_{bra}$, $NPP_{leaf}$ and $NPP_{root}$ are the annual net increments of stems (including bark), branches, leaves, and roots, respectively. Moreover, NPP of aboveground organs ($NPP_{ag}$) is the sum of NPPs from stems, branches, and leaves:

$$NPP_{ag} = NPP_{stem} + NPP_{bra} + NPP_{leaf}, \qquad (3)$$

To simplify our analysis, in this study we did not consider carbon allocations to fruits, flowers and exudates.

### 2.2. Influencing Factors

#### 2.2.1. Soil Data and Stand Characteristics

We collected soil data including alkali-hydrolysable nitrogen (AN, mg/kg), available phosphorus (AP, mg/kg), available potassium (AK, mg/kg), bulk density (BD, g/cm$^3$), soil organic matter (SOM, g/kg) and pH from the China Dataset of Soil Properties for Land Surface Modeling provided by Cold and Arid Regions Sciences Data Center at Lanzhou (http://westdc.westgis.ac.cn/). The data were obtained from the second national soil survey with a resolution of 1 km.

Stand characteristics used in this study involve stand density and age extracted from the literature.

#### 2.2.2. Climatic Variables

Climatic factors used in this study included mean annual temperature (MAT, °C), mean annual precipitation (MAP, mm), mean high temperatures in warm months (HTWM, °C), and mean low temperatures in cold months (LTCM, °C). HTWM and LTCM were used to explore the effects of high and low temperatures on *P. massoniana* NPP, because the accumulation of forest biomass in different study sites is influenced by environmental factors over a relatively long period of time rather than a temporary time [30,31]. For the study area, July and August were warm months, and December was the cold month, as determined by frequency statistics of the months when the annual maximum and minimum temperatures occurred from 1981 to 2010 (Figure 2). Several study sites were located far from meteorological stations, and have different biomass accumulation stages. Therefore, spatially-interpolated climate data within the same observational period (1981–2015) was used. MAT and MAP data (1981–2015) were extracted from a China climate dataset (1 km resolution) provided by the Data Center for Resources and Environmental Sciences, Chinese Academy of Sciences (http://www.resdc.cn). Monthly HTWM and LTCM from 1981 to 2015 were obtained from China National Meteorological Information Center (http://data.cma.cn/) and interpolated into 1 km grid cells employing the kriging method. Spatial data was interpolated with ArcGIS 10.2 software.

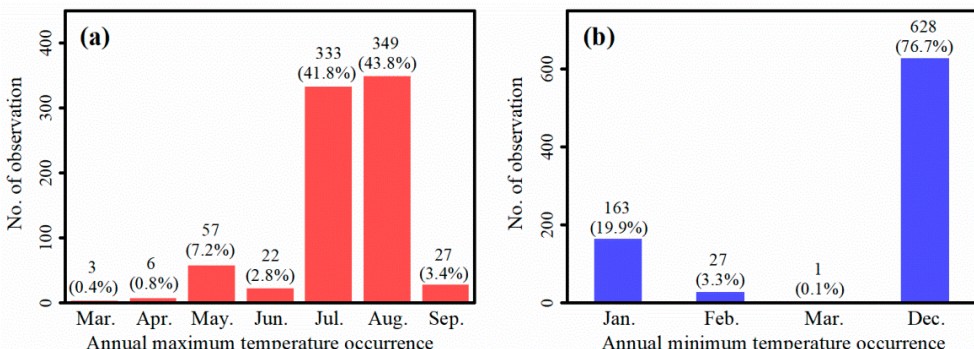

**Figure 2.** Frequency distributions of (**a**) annual maximum temperature and (**b**) annual minimum temperature occurrence time of 819 meteorological stations from 1981 to 2010 in southern China, where *P. massoniana* dominate. Data were obtained from meteorological stations in 11 provinces (Guizhou, Chongqing, Guangxi, Hunan, Fujian, Guangdong, Jiangxi, Hubei, Sichuan, Zhejiang and Anhui) provided by China National Meteorological Information Center (http://data.cma.cn/). The percentage of observations are presented in parentheses.

### 2.3. Statistical Analysis

We performed a descriptive statistical analysis to determine the distribution and variability of the NPP for *P. massoniana* forests, and fitted the distribution and variability curves with Gaussian functions. To stabilize heteroscedasticity and improve model *R*-square values in linear regression,

all variables were transformed using the natural logarithm (i.e., $\ln(x_i)$) prior to analysis. Pearson correlation analyses were used to evaluate the relationship between the NPP of each *P. massoniana* component and site conditions (longitude, latitude, and elevation). The same approach was used to estimate the relationship between environmental factors and site conditions. All statistical analyses above were performed with SPSS (version 20, IBM Corp., Armonk, NY, USA).

The linear mixed-effects models (LMMs) were employed to analyze the effect of environmental factors and stand characteristics on *P. massoniana* NPP. To find the best models for our data we first built a full mixed-effects model containing all of the potential explanatory variables. Based on previous studies [25–27], soil variables (i.e., AN, AP, AK, BD, SOM, and pH), climate variables (i.e., MAT, MAP, HTWM, and LTCM), and stand characteristics (stand age and density), which were not highly correlated were checked by a correlation matrix and selected as potential explanatory variables. The twelve potential explanatory variables were set as fixed-effect terms and study sites were set as random-effect terms, to account for site-specific effects, such as forest management and microsite. Based on the full model, in each analysis, we constructed a set of candidate models that included different combinations of potential explanatory variables, in which we also included the null model with all fixed effects deleted [32]. Models were compared using the Akaike Information Criterion corrected for small samples (AICc). The best-performing model with the lowest AICc was selected as the final model. LMMs were performed using the package "lmerTest" [33] in R software (version 3.6.1) [34]. The model performance was evaluated by the "MuMIn" package [35], in which AICc, marginal *R*-square (variance explained by fixed factors) and conditional *R*-square (variance explained by both fixed and random factors) were calculated [36]. In addition, a regression analysis was performed to analyse the relationships between key environmental factors and NPP based on the results of LMMs.

## 3. Results

### 3.1. Variability of P. mansoniana Distribution and NPP

*P. massoniana* components exhibited large NPP variation across sampling sites (Figure 1, Table 1), ranging from 0.29 to 8.71, 0.10 to 2.11, 0.02 to 2.31, 0.07 to 2.11, 0.88 to 10.81, and 1.04 to 13.13 Mg·ha$^{-1}$·year$^{-1}$ for stems, branches, leaves, roots, aboveground organs and total tree, respectively, with mean values of 3.51, 0.69, 0.34, 0.81, 4.53, 5.65 Mg·ha$^{-1}$·year$^{-1}$, respectively (Figure 3).

**Table 1.** Net primary productivity (NPP) of *P. massoniana* stem, branch, leaf, root, aboveground organs, and total tree. Number of observations (N), mean value (Mean), maximum value (Max), minimum value (Min) and standard error (SE) were reported.

| Component | N | NPP (Mg·ha$^{-1}$·year$^{-1}$) | | | |
|:---:|:---:|:---:|:---:|:---:|:---:|
| | | **Mean** | **Max** | **Min** | **SE** |
| Stem | 172 | 3.51 | 8.71 | 0.29 | 0.13 |
| Branch | 168 | 0.69 | 2.11 | 0.10 | 0.03 |
| Leaf | 168 | 0.34 | 2.31 | 0.02 | 0.02 |
| Root | 148 | 0.81 | 2.11 | 0.07 | 0.03 |
| Aboveground | 185 | 4.53 | 10.81 | 0.88 | 0.15 |
| Total tree | 161 | 5.65 | 13.13 | 1.04 | 0.20 |

In this study, *P. massoniana* geographical distribution covered 11 provinces in southern China and the full geographical distribution of the species was well represented in our database (Figures 1 and 4a). *P. massoniana* study sites ranged from 25 to 1357 m of elevation, and most sample sites were set in the regions below 1000 m (Figure 4b). NPP$_{stem}$, NPP$_{bra}$, NPP$_{leaf}$, NPP$_{root}$, NPP$_{ag}$, and NPP$_{tree}$ showed no clear relationships with longitude and elevation, but significant latitudinal trends ($p < 0.01$, Table 2).

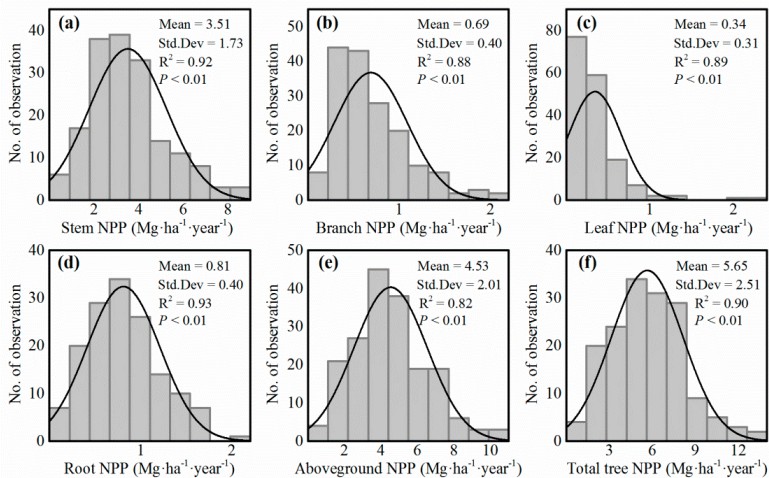

**Figure 3.** Frequency distributions of (**a**) stem, (**b**) branch, (**c**) leaf, (**d**) root, (**e**) aboveground organs and (**f**) total tree net primary productivity (NPP) of *P. massoniana* forests in south China. The mean and standard deviation values are presented. The curves were fitted by a Gaussian function.

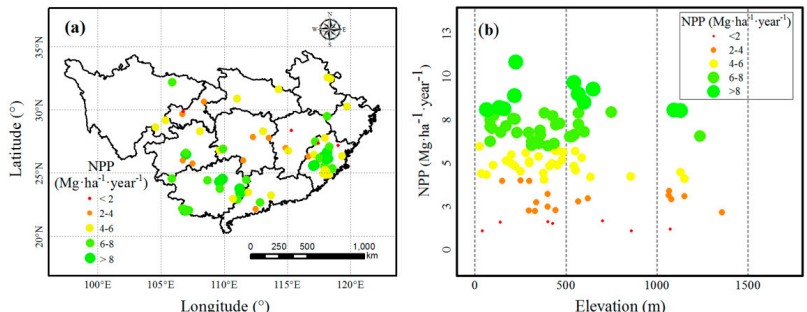

**Figure 4.** Net primary productivity (NPP) spatial distribution of *P. massoniana* forests in relation to longitude, latitude (**a**), and elevation (**b**).

**Table 2.** Pearson correlations between site conditions and *P. massoniana* stem, branch, leaf, root, aboveground, and total tree net primary productivity (NPP) on a logarithmic (ln) scale.

| Site Conditions | $NPP_{stem}$ | $NPP_{bra}$ | $NPP_{leaf}$ | $NPP_{root}$ | $NPP_{ag}$ | $NPP_{tree}$ |
|---|---|---|---|---|---|---|
| Longitude (°E) | −0.112 | −0.054 | −0.007 | −0.110 | −0.100 | −0.119 |
| Latitude (°N) | −0.285 ** | −0.369 ** | −0.208 ** | −0.251 ** | −0.338 ** | −0.344 ** |
| Elevation (m) | −0.087 | −0.019 | 0.102 | −0.069 | −0.061 | −0.094 |

Note: (1) stem net primary productivity ($NPP_{stem}$); branch net primary productivity ($NPP_{bra}$); leaf net primary productivity ($NPP_{leaf}$); root net primary productivity ($NPP_{root}$); aboveground net primary productivity ($NPP_{ag}$); net primary productivity of total tree ($NPP_{tree}$); (2) ** represent $p < 0.01$.

### 3.2. Factors Influencing NPP of Different P. massoniana Components

*P. massoniana* $NPP_{stem}$, $NPP_{bra}$, $NPP_{leaf}$, $NPP_{root}$, $NPP_{ag}$, and $NPP_{tree}$ exhibited different sensitivities to environmental factors and stand characteristics (Table 3). $NPP_{stem}$ was significantly positively correlated with MAT, age, density, AP, SOM and BD, but significantly negatively correlated with HTWM. $NPP_{bra}$ was significantly positively correlated with MAP and density. $NPP_{leaf}$ was significantly positively correlated with MAP, but significantly negatively correlated with age. $NPP_{root}$ was significantly positively correlated with BD, age, and density. $NPP_{ag}$ significantly increased with MAT, age, and density. $NPP_{tree}$ significantly increased with BD, stand age, and density, but significantly decreased with HTWM.

**Table 3.** Summary of the final model to test the effects of environmental factors and stand characteristics on *P. massoniana* stem, branch, leaf, root, aboveground, and total tree net primary productivity (NPP). Only the best-performing model, with the lowest the Akaike Information Criterion corrected for small samples (AICc), is shown (see Table S2 for the results of model performance).

| NPP Component | Parameter | Variable | | | | | | | | | |
|---|---|---|---|---|---|---|---|---|---|---|---|
| | | Intercept | MAT | HTWM | AGE | DENSITY | BD | MAP | AN | AP | SOM |
| Stem | Estimate | 8.797 | 1.575 | −5.099 | 0.562 | 0.216 | 1.170 | — | — | 0.424 | 0.370 |
| | SE | 5.746 | 0.466 | 1.745 | 0.086 | 0.067 | 0.585 | — | — | 0.128 | 0.171 |
| | t-value | 1.531 | 3.378 | −2.922 | 6.520 | 3.233 | 1.999 | — | — | 3.311 | 2.160 |
| | p-value | 0.131 | <0.01 | <0.01 | <0.001 | <0.01 | <0.05 | — | — | <0.01 | <0.05 |
| Branch | Estimate | 2.524 | — | −3.997 | — | 0.148 | 0.913 | 1.294 | — | — | — |
| | SE | 7.119 | — | 2.472 | — | 0.059 | 0.654 | 0.455 | — | — | — |
| | t-value | 0.355 | — | −1.617 | — | 2.533 | 1.397 | 2.844 | — | — | — |
| | p-value | 0.724 | — | 0.111 | — | <0.05 | 0.168 | <0.01 | — | — | — |
| Leaf | Estimate | 6.673 | — | −4.488 | −0.588 | 0.144 | 0.661 | 1.098 | — | — | — |
| | SE | 7.924 | — | 2.777 | 0.097 | 0.073 | 0.725 | 0.515 | — | — | — |
| | t-value | 0.842 | — | −1.616 | −6.045 | 1.964 | 0.912 | 2.133 | — | — | — |
| | p-value | 0.403 | — | 0.111 | <0.001 | 0.052 | 0.366 | <0.05 | — | — | — |
| Root | Estimate | −1.479 | 1.173 | −2.815 | 0.429 | 0.257 | 1.885 | — | 0.833 | — | — |
| | SE | 5.436 | 0.656 | 1.589 | 0.091 | 0.070 | 0.898 | — | 0.439 | — | — |
| | t-value | −0.272 | 1.787 | −1.772 | 4.699 | 3.680 | 2.098 | — | 1.894 | — | — |
| | p-value | 0.786 | 0.079 | 0.081 | <0.001 | <0.001 | <0.05 | — | 0.063 | — | — |
| Aboveground | Estimate | 1.494 | 1.142 | −1.666 | 0.333 | 0.182 | — | — | — | — | — |
| | SE | 3.510 | 0.412 | 0.998 | 0.077 | 0.064 | — | — | — | — | — |
| | t-value | 0.426 | 2.771 | −1.669 | 4.333 | 2.852 | — | — | — | — | — |
| | p-value | 0.671 | <0.01 | 0.099 | <0.001 | <0.01 | — | — | — | — | — |
| Total tree | Estimate | 1.920 | 0.989 | −4.158 | 0.409 | 0.258 | 1.356 | 0.681 | 0.598 | — | — |
| | SE | 4.635 | 0.520 | 1.910 | 0.080 | 0.066 | 0.647 | 0.529 | 0.341 | — | — |
| | t-value | 0.414 | 1.900 | −2.177 | 5.092 | 3.911 | 2.096 | 1.287 | 1.757 | — | — |
| | p-value | 0.680 | 0.062 | <0.05 | <0.001 | <0.001 | <0.05 | 0.202 | 0.083 | — | — |

Note: mean annual temperature (MAT); mean high temperatures in warm months (HTWM); stand age (AGE); stand density (DENSITY); bulk density (BD); mean annual precipitation (MAP); alkali-hydrolysable nitrogen (AN); available phosphorus (AP); soil organic matter (SOM).

### 3.3. Relationship between Site Conditions and Environmental Factors

*P. massoniana* NPPs showed significant trends in latitude, rather than in longitude and elevation (Table 2), therefore the trend of environmental variables in latitude was described minutely here, which was necessary to further explore the cause of formation of the spatial heterogeneity of *P. massoniana* NPP. MAT, LTCM, and MAP were significantly negatively correlated with latitude ($p < 0.01$), with correlation coefficients of −0.291, −0.413, and −0.256 on a logarithmic scale, respectively (Table 4). HTWM showed no clear relationship with latitude (Table 4). For soil properties, SOM was significantly negatively correlated with latitude ($p < 0.01$), with a correlation coefficient of −0.216 on a logarithmic scale (Table 5). AP and PH were significantly positively correlated with latitude ($p < 0.01$), with correlation coefficients of 0.223 and 0.216 on a logarithmic scale, respectively (Table 5). AN, AK, and BD showed no clear relationship with latitude (Table 5).

**Table 4.** Pearson correlations between site conditions and climate factors on a logarithmic (ln) scale.

| Variables | MAT | LTCM | HTWM | MAP |
|---|---|---|---|---|
| Longitude (°E) | −0.073 | −0.173 * | 0.352 ** | 0.242 ** |
| Latitude (°N) | −0.291 ** | −0.413 ** | −0.061 | −0.256 ** |
| Elevation (m) | −0.334 ** | 0.319 ** | −0.391 ** | −0.048 |

Note: (1) mean annual temperature (MAT); mean low temperatures in cold months (LTCM); mean high temperatures in warm months (HTWM); mean annual precipitation (MAP). (2) * and ** represent $p < 0.05$ and $p < 0.01$, respectively.

**Table 5.** Pearson correlations between site conditions and soil factors on a logarithmic (ln) scale.

| Variables | AP | AN | AK | PH | SOM | BD |
|---|---|---|---|---|---|---|
| Longitude (°E) | −0.068 | −0.175 * | −0.291 ** | 0.051 | −0.341 ** | 0.152 * |
| Latitude (°N) | 0.223 ** | −0.093 | −0.074 | 0.216 ** | −0.216 ** | −0.112 |
| Elevation (m) | 0.016 | 0.417 ** | 0.292 ** | 0.153 * | 0.280 ** | −0.213 ** |

Note: (1) bulk density (BD); alkali-hydrolysable nitrogen (AN); available phosphorus (AP); soil organic matter (SOM); available potassium (AK). (2) * and ** represent $p < 0.05$ and $p < 0.01$, respectively.

## 3.4. Relationship between Temperatures and P. massoniana NPP

*P. massoniana* NPP correlated highly with temperatures. As the MAT increased, NPP of both *P. massoniana* components (i.e., $NPP_{stem}$, $NPP_{bra}$, $NPP_{leaf}$, $NPP_{root}$, $NPP_{ag}$, and $NPP_{tree}$) increased signifcantly ($p < 0.05$, Figure 5a,d,g,j,m,p). In addition, LTCM was also significantly positively correlated with $NPP_{stem}$, $NPP_{bra}$, $NPP_{leaf}$, $NPP_{ag}$, and $NPP_{tree}$ ($p < 0.05$, Figure 5b,e,h,n,q).

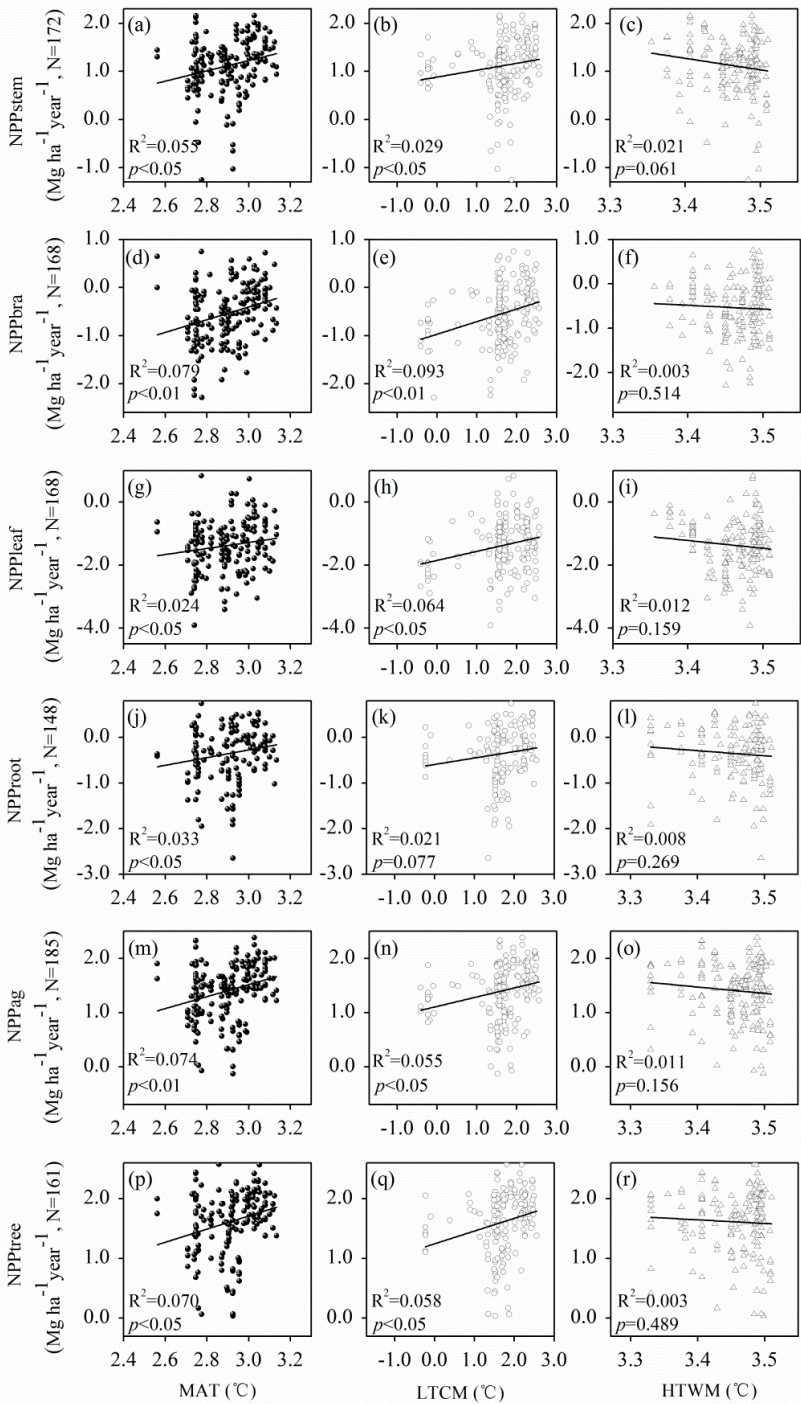

**Figure 5.** The relationships between *P. massoniana* net primary productivity (NPP) and MAT (**a,d,g,j,m,p**), LTCM (**b,e,h,k,n,q**), and HTWM (**c,f,i,l,o,r**) on a logarithmic (ln) scale. Mean annual temperature (MAT); mean low temperatures in cold months (LTCM); mean high temperatures in warm months (HTWM).

## 4. Discussion

### 4.1. Factors Influencing NPP of P. massoniana Forests

#### 4.1.1. Climate Effects

*P. massoniana* $NPP_{stem}$ and $NPP_{ag}$ were significantly positively correlated with MAT (Table 3). This finding is consistent with previous studies [37–39], indicating that the warming effect can promote biomass accumulation of *P. massoniana*. Zhang et al. (2011) [30] showed that low air temperature from January to March was the major factor controlling the interannual variations of net carbon uptake, and cold springs greatly shortened the growing season length, reducing the carbon uptake period. Similarly, Zheng et al. (2012) [31] found that even a short-term freezing event during the transitional periods from dormancy to growth in early spring could result in disastrous damage to pine forests in temperate China. However, HTWM was significantly negatively correlated with $NPP_{stem}$ and $NPP_{tree}$ (Table 3), because higher temperatures in summer can increase plant evapotranspiration and respiration rates, leading to decreased biomass accumulation rates in subtropical forests [40]. Therefore, temperatures had an important effect on *P. massoniana* NPP. Large differences of NPP in sensitivity to temperature and precipitation were identified among various biomes and regions [41,42]. Regression analysis showed that NPP of both *P. massoniana* components increased significantly as MAT increased (Figure 5a,d,g,j,m,p). LMM results showed that MAT was positively correlated with $NPP_{stem}$ and $NPP_{ag}$ (Table 3), which formed relatively high proportions of $NPP_{tree}$ (Table 1). Although $NPP_{bra}$ and $NPP_{leaf}$ were also positively correlated with MAP (Table 3), NPP of these two components only formed relatively low proportions of $NPP_{tree}$ (Table 1). In fact, tree growth in our study region is probably not often limited by precipitation. Moreover, the magnitude of variation in MAP was not large enough to lead to significant changes in forest productivity in this region, since *P. massoniana* is distributed in a large gradient zone of precipitation in south-east Yangtze River basin and the southeast rivers [43]. Liang et al. (2015) [44] found that air temperature was the dominant climatic factor that controlled the interannual variability in NPP throughout the country except for arid and semi-arid regions in the middle–north and northwest parts of China. Similarly, Wang et al. (2011) [45] found that forest NPP appeared to be primarily a function of temperature, not precipitation, in subtropical Louisiana, USA. Furthermore, Churkina and Running (1998) [46] reported that temperature appeared to be the primary control on evergreen needle-leaved forest NPP on a global scale. Overall, our findings were in line with previous studies, indicating that *P. massoniana* NPP was more sensitive to temperature than to precipitation.

#### 4.1.2. Soil Effects

*P. massoniana* $NPP_{stem}$ significantly increased with SOM and AP (Table 3), indicating that, although this species can be used as a pioneer species for afforestation in infertile soil, soil with sufficient nutrients can promote biomass accumulation of *P. massoniana*. AP was significantly positively correlated with $NPP_{stem}$, while AN and AK were not significantly correlated with *P. massoniana* NPP (Table 3). NPP of *P. massoniana* forests was more sensitive to soil AP than to soil AN and AK. This result may be attributed to the fact that southern China is dominated by acidic red soil, where phosphorus is mostly fixed in insoluble form, and the AP content directly absorbed by plants is extremely low [47]. Therefore, the response of *P. massoniana* to phosphorus deficiency has received extensive attention in the context of N deposition in recent years [48]. In our study, soil BD was significantly positively correlated with $NPP_{stem}$, $NPP_{root}$, and $NPP_{tree}$ (Table 3). However, there were relatively few studies on the relationship between soil physical properties and tree growth of *P. massoniana*. Therefore, this result needs to be further confirmed.

### 4.1.3. Stand Characteristics Effects

During this study, *P. massoniana* $NPP_{stem}$ significantly increased, whereas $NPP_{leaf}$ significantly decreased with age (Table 3). This finding can be explained by the variation in biomass allocation patterns with tree growth. Stems, as support tissues, obtain more biomass investment with tree growth, at the expense of branch and leaf biomass [49]. Wood (dead cells) continuously accumulates in the stem as trees grow, whereas branch and leaf biomass decrease as the early death of lower branches is caused by mutual shading [49]. Additionally, following canopy closure, older and taller trees entail greater expenditures for their construction and maintenance and expend more energy than younger and shorter trees to supply leaves with the same amount of water [49]. *P. massoniana* $NPP_{stem}$, $NPP_{root}$, $NPP_{ag}$, and $NPP_{tree}$ significantly increased with age (Table 3), probably because the forest stands in our database were mostly middle-aged plantations (Table S1), in which age-related decline in forest productivity has not yet occurred [50,51]. Stand density was significantly positively correlated with $NPP_{stem}$, $NPP_{bra}$, $NPP_{root}$, $NPP_{ag}$, and $NPP_{tree}$ (Table 3), indicating that density had a marked impact on productivity estimation. This finding was similar to that of Bormann and Gordon (1984) [52], who found that aboveground mass and total mass per unit area were the greatest in the dense stands. In contrast to the trends in per-unit-area values, for average tree values, dry mass of all individual-tree components and totals were the highest in the open stands, and the lowest in the dense stands [52]. At lower stand densities, less growth per unit area is obtained, but this will be offset by greater growth per tree [53]. In a 42-year thinning experiment, Horner et al. (2010) [54] found that moderately thinned stands had the highest aboveground carbon storage rate and the highest aboveground carbon stocks rather than unthinned treatment (highest density). Therefore, although there was a significant positive correlation between NPP and density in this study, this relationship varied between stands due to the trade-off between maximizing individual tree size or stand yield [53]. Therefore, for long-term managed plantations, the initial density, thinning practice and the competition–density effect should be considered in productivity predictions [55,56].

### 4.2. Latitudinal Effects on NPP of P. massoniana Forests

In this study, *P. massoniana* $NPP_{stem}$, $NPP_{bra}$, $NPP_{leaf}$, $NPP_{root}$, $NPP_{ag}$, and $NPP_{tree}$ exhibited inverse latitudinal trends, but no longitudinal and elevation trends (Table 2). Liu et al. (2016) [26] found that forest NPP show no clear relationship with longitude but negatively correlate with latitude in southwestern China. Ni (2003) [11] also concluded that forest NPP significantly decreased with increasing latitude based on forest inventory data in China between 1989 and 1993. In the study of Zhan et al. (2018) [27] on forests in eastern China, the functional relationship between NPP and latitude was: $y = 0.06x^2 - 3.91x + 73.71$ ($R^2 = 0.38$, $p < 0.001$), NPP decreased first and then increased with the increase in latitude, and monotonically decreased with increasing latitude when the latitude was below 35.0 °N. Thus, the results of our study are consistent with previous studies.

Latitude is not an environmental factor capable of having direct influence on *P. massoniana* growth [57,58], but rather an indirect variable that condenses a set of factors that vary from south to north, including MAT, LTCM, MAP, AP, PH, and SOM in this study (Tables 4 and 5). Climatically, the north-to-south and west-to-east gradients in China both reflect shifts from cold and dry to warm and moist conditions, although the thermal gradient is steeper in the former and the moisture gradient more pronounced in the latter [25,59]. The correlation analysis of latitude and climate indicators showed that MAT, LTCM, and MAP decrease with increasing latitude (Table 4), which is in line with the generally accepted idea that the southern part of China is warmer and moister than the northern part. It is worth noting that HTWM, which was significantly negatively correlated with *P. massoniana* $NPP_{stem}$ and $NPP_{tree}$ (Table 3), did not show a significant latitudinal trend (Table 4). This suggested that high temperatures were a significant climate stressor across the species geographical distribution and were not restricted to marginal populations at low latitudes. MAT and MAP, which were significantly positively correlated with NPP (Table 3), showed inverse latitudinal trends (Table 4). Therefore, both temperature and precipitation contributed to the formation of the inverse latitudinal trends of

*P. massoniana* NPP. Temperature was a key contributing factor to these trends, as NPP was highly sensitive to any change in temperature. AP and SOM, which were significantly positively correlated with *P. massoniana* productivity (Table 3), showed different latitudinal trends (Table 5). Latitude was significantly positively correlated with AP, but significantly negatively correlated with SOM (Table 5). In fact, low phosphorus availability is a limiting factor on *P. massoniana* growth not only at low latitudes, but also in the broad red soil areas in southern China [60]. In summary, the latitudinal trends of *P. massoniana* productivity are driven by the significant influences of climate and soil properties.

*4.3. Uncertainty Analysis*

In this study, some uncertainties in forest productivity assessment have been eliminated in the process of developing the biomass database. However, there was still a great deal of variability in influencing factors. Atmospheric N deposition, elevated $CO_2$ concentration, and climate warming have been confirmed to affect *P. massoniana* growth [61,62]. Moreover, they can affect other biological processes and thus indirectly affect forest productivity. For example, N deposition will further aggravate soil acidification and change soil nutrient conditions (e.g., N:P ratio) [63,64]. Furthermore, in the process of biomass data acquisition, it was found that the size classes for *P. massoniana* roots were not uniform across studies, which made it difficult to establish a complete database containing root sizes, although roots with diameter < 2 mm were usually defined as fine roots. Pan et al. (2018) [65] suggested that understory vegetation generated different effects on soil carbon and nitrogen processes in aerially seeded *P. massoniana* plantations. Therefore, the missing components in field measurements, including shrubs, herbs, and litterfall, not only lead to an underestimation of actual NPP, but their significant contribution to biological processes of tree growth should also be considered in future studies.

## 5. Conclusions

We established a new regional database of *P. massoniana* forest biomass. Tree components (stem, branch, leaf, root, aboveground organs and total tree) exhibited large NPP variation across sampling sites. Climate, soil, and stand characteristics have a marked impact on the NPP of *P. massoniana* forests, and the influence of these variables on the NPP of different tree components varied. Appropriate temperature and soil nutrients, especially soil AP, are beneficial to *P. massoniana* growth. NPP of all tree components of *P. massoniana* exhibited inverse latitudinal trends, which were driven by the significant influences of climate and soil properties. Temperature was a key environmental factor for the formation of these trends. Future studies should place a particular emphasis on the effects of various components of the forest ecosystem, particularly soil, litterfall, and understory vegetation on the biological processes of tree growth, and their interactions with climate change.

**Supplementary Materials:** The following are available online at http://www.mdpi.com/1999-4907/11/4/404/s1, Table S1: Primary data of site condition, plot size, stand age, density, component biomass (stem, B1; branch, B2; leaf, B3; root, B4; aboveground organs, B5; and total tree, B6) and measurement year for *Pinus massoniana* forests in southern China. Table S2: Parameters fitted to the final model, full model, and null model by means of linear mixed-effects models.

**Author Contributions:** Conceptualization, X.H. and M.T.; methodology, Z.Z. and P.W.; formal analysis, P.W.; writing—original draft preparation, X.H.; writing—review and editing, Z.Z.; visualization, X.H. and C.H.; funding acquisition, M.T. All authors have read and agree to the published version of the manuscript.

**Funding:** This research was funded by the National Key R&D Program of China (2016YFD0600201), the Major Scientific and Technical Innovation Project of Hubei Province, China (2018ABA074) and research platform support was provided by the Long-Term Track Research Program of the Forest Ecological Station in the Three Gorges Reservoir Region (Zigui) of the Yangtze River, China.

**Acknowledgments:** The authors gratefully acknowledge financial support from the National Key R&D Program of China (2016YFD0600201), the Major Scientific and Technical Innovation Project of Hubei Province, China (2018ABA074) and research platform support from the Long-Term Track Research Program of the Forest Ecological Station in the Three Gorges Reservoir Region (Zigui) of the Yangtze River, China.

**Conflicts of Interest:** The authors declare no conflict of interest.

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
