# Peer review of "Net Primary Productivity of Pinus massoniana Dependence on Climate, Soil and Forest Characteristics"

_forests, doi:10.3390/f11040404_

Round 1

Reviewer 1 Report

This paper develops a biomass database for P. massoniana in southern China by collecting data from the literature, and then calculates net primary productivity (NPP) and explores its relationships with climatic and geographic variables. Though the manuscript was quite well prepared, it should be significantly improved before publication.

The analysis steps conducted in this paper are generally simple. Authors just developed a master table of biomass data from published work and explored the distribution of data. Statistical models between NPP and other variables were not described in advance. I think this paper can be significantly improved by:

- Focus on accuracy assessments. How can you evaluate the quality of a dataset combining data from various sources? Biomass data can be measured using different methods with different accuracy levels. In line 366, you said some uncertainty assessments have been done but I could not find any such analysis throughout the paper.

- Improve the descriptions of methods used for exploring the relationships between NPP and environmental factors.

Line 129-130: I am not convinced by the approach used for deriving latitude and longitude. How did you get exact Lat and Long numbers by site name on Google Earth? Something like site name is very general information, it does not tell you the exact location of a plot. Even this method also used in Xiaoyun et al. 2018 (as your reference), I personally do not agree. The data table (Table S1): where available, please include plot sizes and measurement year for each record. These will be very useful for the forest community. You should have measurement year as you need it for calculating NPP???

Also, I would reckon this table should be sorted by: first site name then measurement time, rather than references.

Please move Tables 4 and 5, and Figure 5 and associated descriptions to the Results section. It is not a consistent structure when presenting some new results in the Discussion. Make Figure 5 larger. Figure 1: Remove the white box on the right of the figure. What do you mean by showing that redundant box? The abstract should contain some numerical results. Line 124: What about sites impacted by natural disturbances? Few grammatical errors, please check carefully.

Reviewer 2 Report

In the presented manuscript Huang et al. concerned very interesting issue – spatial variation of forest productivity. This topic was always very attractive to me, in terms of models applicability and simplification of biomass calculation. Revealing geographic patterns of productivity will allow future improvements of global carbon budget estimations and closing gaps in measurements. For that reason I appreciate the topic of the undertaken study. Data collected by the Authors seems to be reliable and sufficient for realization of the study aim. Protocol of literature review is clear, well-documented and allows replicability.

My main concern is about data analysis. Authors used the simplest possible method – multiple linear regression, assuming distributions normality and linear relationships between variables, as well as lack of intercorrelations. This method also assumes the independence of observations.  Here Authors used multiple observations from the same study or even study site. Unknown dependencies (e.g. similar type of forest management, microsite, methodology of sampling, accuracy of measurements – all which we have no influence on) violate the assumption of sample independence in this case. This may be easily accounted by mixed-effects models: Authors should account for study site and study as random effects in the models developed. Another problem is intercorrelation: including highly correlated (rule of thumb says that r>0.7) variables in model may increase R2, but lead to variance inflation. One might think that adding a particular variable increases information used by model, but this is the same information. For example, including stand volume and height results the same information, as well as stand age and height.

Authors provided mere coefficients of final models. They did not show us SE of particular variables, nor test statistics. We are not sure how models were developed? Did Authors tested randomly selected combinations of variables aiming in maximalization of adjusted R2, or did Authors started with global model (i.e. including all relevant variables)? How variables were chosen? How Authors ensured that variables are not intercorrelated, which lead to variance inflation and increasing R2? How Authors ensured about models parsimony (i.e. the simplest of the best)? I recommend Authors to first check correlations between variables using correlation matrix or variance inflation factors, then choose only variables which are not intercorrelated, build global model (including all relevant variables) and reduce the models to decrease model AIC (Akaike’s Information Criterion). AIC will allow to include only relevant variables, as AIC penalize for additional terms in model which not provide significant increase of explained variance. In this way AIC not allow us to build models with too much variables. I expect table with models with following columns: NPP component, variable, estimate, SE, test statistic, p. For models I also expect providing AIC of full model, of null model (intercept-only, it will show us how much final model is better than imputing mean value to all observations) and AIC of final model. Authors may also provide marginal and conditional R2 for these models – marginal R2 tells us how much of variability is explained by fixed-effects only (i.e. these variables which we study) and conditional – by both fixed and random effects (i.e. includes all effects connected with samples dependency).  

My another objection is connected with light – although Authors used this word in abstract and conclusions describing important factors determining NPP of the studied species, there are no variables describing properly light availability. Solar radiation express the amount of solar energy reaching the particular area, but this variable has low information about light quantity available for plants. In the scale of study area better variable would be photoperiod, which might influence biomass allocation (Körner C, Basler D. 2010. Phenology under global warming. Science 327: 1461–1462.) Another possible solution (and simpler) will be not accounting for light in this study and focus on soil and climatic variables – as more relevant in this spatial scale (Pearson and Dawson 2003. Glob. Ecol. Biogeogr. 12:361-371).

Literature include mainly publications of China (37/51) – I suggest to refer to studies from other parts of the world, e.g. Schepaschenko et al. (2018). Improved Estimates of Biomass Expansion Factors for Russian Forests. Forests or Poorter et al. (2012) Biomass allocation to leaves, stems and roots: Meta-analyses of interspecific variation and environmental control. New Phytologist.

As data analyses needs revision, I will detailly comment only introduction and methods, and relevant parts of abstract.

Specific comments:

Title – it is too long and repetitive: biogeographical patterns of variation with climate, soil and forest characteristics. Biogeographical patterns also concerns climatic variability. In my opinion it would be better to change it e.g. to “”

Abstract

18 – please provide authorities of botanical name with first usage (first in abstract and first in the main text)

19 – ‘enormous’ – maybe better use word ‘high’, which is more neutral and less exaggerated

18 – ‘has brought’ – suggests that this species was earlier absent in the study area. This species is native, thus please rephrease

19 – was making a database the main aim of the study?

20 – what is tree layer? not clear without reading the main text. Please change into aboveground biomass

29 – what is MWHT? explanation is not clear – better use name from main text (mean high temperatures in warm months)

Introduction

Authors once describe carbon, once biomass, once NPP – please state more clear how it is linked and how knowledge about NPP contribute to better carbon cycling understanding

72-84 – please state clearly that P. massoniana is native species – this is an essential information for people assessing impacts of plantations, which can automatically assess plantation = alien, which is not true.

99-106 – I appreciate the aim of Authors to make data checked, synthetized and available. This will be also good for a paper and journal in terms of citability.

103 – “to Western scientists” – maybe better to ‘non-Chinese’? I am sure that the same problem will have Western (assumable English-speaking), as Southern (e.g. Thai or Wietnamese), Eastern (Japan) and Northern (Russian), as well as in Latin America or other countries.

107-115 – Aims 2 and 3 are more or less similar – longitude, latitude and elevation are also environmental factors, and has no strict effect on plants. Tree does not know its geographic coordinates, but responds to climate. For that reason coordinates are proxies of climate (and – indirectly – of soil conditions, as soil is climate-dependent). Maybe it would be interesting to develop models based on environmental variables only and coordinates only and compare their performance?

Methods

155-160 – How Authors ensured that such variables are not intercorrelated?

162-167 – why such variables were chosen? Please provide the references stating that such variables might be relevant to biomass. I also recommend using more widely used climatic variables, to increase applicability of results and comparison with other studies. Variables from bioclim (https://www.worldclim.org/bioclim) will be better.

167-168 – what are ‘different biomass accumulation stages’?

182 – there is no need to make a separate subchapter for single sentence. Please incorporate it into other parts, e.g. join with soil data

Results

Descriptive part should be concentrated (3.1 and 3.2 joint) and entitled ‘Variability of P. mansoniana distribution and NPP’.

Round 2

Reviewer 1 Report

I am not satisfied with some of the author’s responses and the revised version.  

1st round comment in normal texts, 2nd round comment in capital texts.

- "The analysis steps conducted in this paper are generally simple. Authors just developed a master table of biomass data from published work and explored the distribution of data. Statistical models between NPP and other variables were not described in advance. I think this paper can be significantly improved by:

Focus on accuracy assessments. How can you evaluate the quality of a dataset combining data from various sources? Biomass data can be measured using different methods with different accuracy levels. In line 366, you said some uncertainty assessments have been done but I could not find any such analysis throughout the paper."

WHAT I MEAN HERE IS THAT THE AUTHORS SHOULD IMPROVE THE QUALITY OF THE PAPER BY FOCUSING ON QUALITY ASSESSMENT OF THE BIOMASS DATASET, NOT JUST EXPLAINING HOW BIOMASS CAN BE MEASURED IN DIFFERENT WAYS (POINT 1).

Improve the descriptions of methods used for exploring the relationships between NPP and environmental factors.

NO RESPONSE OR IMPROVEMENT MADE FOR THIS POINT.

- Line 129-130: I am not convinced by the approach used for deriving latitude and longitude. How did you get exact Lat and Long numbers by site name on Google Earth? Something like site name is very general information, it does not tell you the exact location of a plot. Even this method also used in Xiaoyun et al. 2018 (as your reference), I personally do not agree.

Author’s response: “For this method, we have two explanations. First, almost all the original references provide accurate sample locations, deriving latitude and longitude was merely used in a few sample sites. Second, this study was conducted at a large regional scale (southern China) rather than a landscape or stand scale. Therefore, although there is a slight bias in the process of obtaining latitude and longitude based on site name, this method may be relatively effective when the research is carried out at a large spatial scale.”

I AM NOT CONVINCED WITH THIS RESPONSE AS:

THOUGH THIS STUDY CONDUCTED OVER A LARGE AREA, IT WAS DONE USING A SAMPLE-BASED ANALYSIS. “A SLIGHT BIAS” NEED TO BUT SOMEHOW IDENTIFIED OR QUANTIFIED.

I THINK REMOVING THE POINTS WITHOUT LAT & LONG DATA (AS YOU SAID, ONLY A FEW SITES) WILL IMPROVE THE QUALITY OF THE DATASET

Reviewer 2 Report

Authors thoroughly corrected the manuscript and detailly responded to all issues raised. However, some parts of explanation included in response should be incorporated to the manuscript, especially about variables selection and variables intercorrelation. It is especially easy, as the text is ready to include in the manuscript. Although Authors response is clear for me, Readers have no access to it, therefore this must be added to the text.

Authors covered almost all of my methodological concerns, but the one was omitted – assumption about observations independence. If Authors not added random effects to the model, they could at least acknowledge that this might influence results – for example by one or two sentences in methods section near data sources descriptions.

I apologize for lack of changed tittle proposal – this was ‘Net primary productivity of Pinus massoniana dependence on climate, soil and forest characteristics’ – shorter and not suggesting local scope of the study, as Authors covered most of its native range.  

Specific comments:

Table 3 – p=0.0000 should be changed into p<0.0001
